# Impacted Canine Management Using Aligners Supported by Orthodontic Temporary Anchorage Devices

**DOI:** 10.3390/ijerph20010131

**Published:** 2022-12-22

**Authors:** Mario Greco, Monika Machoy

**Affiliations:** 1Postgraduate School of Orthodontics, University of Ferrara, Via Fossato di Mortara, 44100 Ferrara, Italy; 2Department of Periodontology, Pomeranian Medical University, ul. Powstańców Wielkopolskich 72, 70-111 Szczecin, Poland

**Keywords:** impacted teeth, TADs, aligners

## Abstract

Introduction: Temporary anchorage devices (TADs) represent an essential instrument under difficult anchorage conditions, especially when the procedure is approached with an aligner technique. The objective of this paper is to describe a possible sequence of orthodontic treatment of impacted canines with aligners supported by orthodontic mini-screws. Materials and Methods: The resolution of impacted canines requires a specific clinical sequence constituted by different steps: the space creation, the surgical exposure, and the orthodontic traction (on the horizontal, vertical, and buccal direction). Following this sequence, two different clinical scenarios can be identified following the space available and the initial malocclusion. The first scenario is constituted by recreating the space for the impacted canine along with the correction of the malocclusion by means of an aligner system and then approaching the de-impaction with TADs. The second clinical scenario is related to the canine-first approach, an immediate de-impaction stage based solely on the use of TADs and sectional wires, and then a finishing phase with aligners. Results: Both approaches to the treatment of impacted canines can be considered reliable, but of course selecting one or the other depends on the space available in the upper arch and on the initial malocclusion. Conclusions: The use of aligners in the treatment of impacted canines in combination with TADs and sectional wires represents a viable alternative option to the conventional systems for canine disinclusion. When the treatment is managed with the presented approaches, no further cooperation with the patient is required in order to support the forced eruption, and an ideal biomechanical approach can be easily applied with one or two mini-screws.

## 1. Introduction

The inclusion of canines following the third molars occurs most frequently, with an incidence ranging from 1% to 5.9% [1]. The most common causes of canine retention described in the literature include disturbances in the position of the tooth bud, lack of space, abnormal eruption path, supernumerary teeth, and genetic factors.

The retained upper canine may manifest itself as palpable by the patient and by the doctor’s palpation of the mucosa in the atrium or on the palate. The proximity of the impacted upper canine may influence the position of the lateral incisor. The palatal canine can press against the root of the lateral incisor, which then can move in the direction given by the canine. The vestibular position of the canine may cause a positive deflection of the crown of the lateral incisor [2]. Since mesioclination is characteristic of most retained canines, they influence the position of the lateral incisor much more often than the first premolar. A disorder observed in the presence of a retained upper canine is also the presence of root resorption of adjacent teeth. Ericson and Kurol define the frequency of its occurrence within 12% [3,4]. They also describe cases of aplasia and other disorders of the morphological structure of the tissues of the lateral incisor, or in its absence, in the case of an impacted upper canine. Kettle, on the other hand, observed a loss of vitality and increased mobility of permanent incisors in the case of the adjacent impacted upper canine [5]. Assessment of the topography of the retained upper canine requires an X-ray. For this purpose, intraoral, occlusal, and oblique jaws, or telerentgenographic and, above all, pantomographic pictures are taken; listed before, the pantomographic ones are an auxiliary material [6,7,8].

As a result, several orthodontic techniques, including specific devices and surgical approaches, have been proposed to track an impacted canine into the arch. Most techniques involve using fixed appliances and a number of auxiliaries in order to favour forced eruption in the crestal bone. In the last few years, a growing demand for orthodontic treatment with removable aesthetic appliances as aligners can be noticed, especially for the adult population. The development of aligner techniques enables to treat more and more difficult and advanced malocclusions along with orthognathic defects [9]. However, the treatment of impacted canines with aligner techniques, besides obvious advantages such as aesthetics and facilitated hygiene, represents a challenge and requires auxiliary devices in order to obtain stable and safe anchorage and to avoid a high level of patient compliance in connection with the surgical procedure.

In fact, as indicated in the literature [10], in order to exert an extrusive force, a combination of elastics and buttons along with aligners should be applied. It means that a higher level of compliance is required from the patient. Moreover, in some specific cases of canine inclusion, the use of leading elastics can be limited, since this option is creating an exclusive vertical vector of eruption, and sometimes a horizontal distal vector followed by a buccal vertical vector is needed instead. Moreover, in case of palatal impaction, in order to use leading elastics, a surgical open eruption should be followed with consequent periodontal problems.

In order to overcome these limitations and to apply an ideal biomechanical approach, TADs (temporary anchorage devices) can be combined with aligners following a hybrid consecutive approach. Several clinical applications of TADs can be managed by palatal insertion [11], but when canines are palatally impacted, only limited places in the bone can be considered for the creation of a skeletal anchorage system. A CBCT analysis is always required for the insertion of palatal TADs, together with a 3D design of the appliance developed in the lab [12]. A reliable and easier alternative to the previous approach could be a combination of aligners and direct TADs, which do not require a mandatory CBCT and a laboratory passage to create the transpalatal bar for vertical anchorage. The ideal ectopic canine sequence requires different steps:space creation in the arch for permanent canines with a preliminary aligner phase;surgical exposure following an infracrestal guided closed eruption to avoid periodontal issues;TAD insertion and, in the case of adult patients, bone modulation along the canine path;deimpaction traction (horizontal, vertical, and buccal depends on the canine position) and an ideal biomechanical system based on the combination of TADs and sectional cantilevers.Torque finishing with aligners.In cases where the space for eruption is already available in the dental arch, it is possible to avoid the first stage and start immediately with the deimpaction system.

## 2. Materials and Methods

This report aims to describe two possible clinical scenarios normally encountered when impacted canines are approached by means of a clear aligner system.

The first scenario involves recreating the space for the impacted canine by means of aligners, along with the correction of the malocclusion. A consequent hybrid approach, combining the use of TADs and sectional wires for the canine de-impaction following the ideal traction, is applied as soon as the space is available. Additional aligners are used again to refine the canine position when the crown of the impacted tooth is pretty well accessible.

The second clinical scenario is related to the canine-first approach, an immediate de-impaction stage based only on the use of TADs and sectional wires. When the canine forced eruption phase is completed, a final clear aligner stage is used to refine the canine torque and position.

### 2.1. Case One—Space Creation for the Impacted Canine and Malocclusion Correction by Means of Clear Aligner System First and Then Consequent Hybrid Treatment with TAD and Sectional Wires

A 16-year-old male patient presented with a Class I occlusion, mild crowding, deep bite, and missing upper left cuspid with the absence of the respective deciduous. Light crowding can be detected in the maxillary arch, whereas no crowding was detected in the mandibular arch. The maxillary midline was coincident with the face and the mandibular midline. The radiographic analysis revealed a moderate impaction with a supernumerary tooth in the lower jaw between 3.5 and 3.4 ligually positioned and a skeletal Class I malocclusion with deep bite due to negative upper and lower incisors inclination.

Figure 1 shows an initial stage of treatment—the upper arch with the absence deciduous canine no. 63. The malposition of the teeth, such as distorotation of the incisors, vestibuloposition of the right canine, as well as an irregular and incorrect shape of the upper arch, are noticeable.

Figure 2 panoramic and CBCT showing the moderate palatal 2.3 impaction. Lateral X-rays confirming a Class I skeletal and dental malocclusion with deep bite.

### 2.2. Treatment Objectives

Treatment objectives were to:

1: align and level the arch exerting a torque correction for upper and lower incisors in order to correct the deep bite and to gain space in the arch for the impacted canine;

2: recover the impacted canine;

3: refine alignment and occlusion.

#### 2.2.1. Aligning and Levelling the Arch

In order to align the teeth and create enough space for de-impaction, the first aligner stage consistent of 25 aligners was necessary. A ClinCheck→(AlignTechnology, San Jose, CA, USA) (a 3-dimensional virtual representation of a doctor’s prescribed treatment plan) reflected the stages of treatment was used. The program enables to measure exactly how many millimetres of space are necessary, using, as a reference, the opposite canine diameter and adding 2–3 mm extra for the bonding buttons and traction system. A total of 2 mm of expansion per quadrant was planned. Moreover, a proper leveling of the lower curve of Spee by means of bite ramps and torque expression of upper incisors has been planned.

In order to put precise forces to the teeth adjacent to the canine, the attachments on the lateral and first premolars were prescribed. This procedure allows also to avoid aligner fitting problems. The result of the first treatment stage (aligned teeth, expanded arch and increased space for permanent canine) is presented in Figure 3.

#### 2.2.2. Recover Impacted Canine

Recovering of the impacted canine was possible using TAD. The ideal position for the TAD insertion is strictly related to the canine impaction position into the bone. The bone condition could affect TAD stability, and this is the reason why no bone drilling is performed. There is only drill free insertion between the roots on the labial or palatal side. Statistically, canines are impacted more frequently on the palatal side of the arch. The palate represents an area considered an anatomically safe zone for insertion, in particular between the roots of the second premolar and the first molar. In order to enhance canine de-impaction in adult patients, some auxiliary surgical procedures such as corticotomy with piezosurgery or bone perforations could be planned along the eruption path of the canine.

##### De-Impaction Traction and Biomechanical System

The canine position in the bone is important for both the TAD positioning and surgical procedure. When the impacted canine is placed horizontally or near the horizontal plane and is close to the adjacent teeth, the best decision is to start with distal traction with a lower vertical component in order to move it far from the other teeth. It is possible to change the traction in a more vertical vector only when the canine is visible.

When the impacted canine is in its vertical position, it is possible to start immediately with vertical traction using a hybrid system combining TADs and sectional wires.

Figure 4 shows the first TAD approach for canine traction two mini-screws (Spider Pin 1.3 × 10 mm, HDC, Sarcedo, Italy) have been inserted. The sectional wire directly bonded on the TADs is a 0.18 SS Australian wire.

The applied biomechanical system includes the type of anchorage that will be attached to the TADs:

• TAD Direct Anchorage, when possible, is the preferable option, since the collateral effects and detrimental forces deriving from canine traction dissipate on mini-screws, thus avoiding undesired movements of the adjacent teeth. The force on the TAD could lay on the same plane when the canine is palatal or crossing from palatal to buccal when a more labial position is needed.

• TAD Indirect Anchorage is not a preferable option, since it means that the traction of the impacted canine could generate collateral forces on the adjacent teeth that need to be connected to the mini-screws to cope with these reactional forces. However, in this situation, in order to avoid changing the TAD position, an indirect anchorage fixing the position of the 2.5 has been created by means of a 0.18 SS Australian wire bonded on the TADs and on the 2.5. In order to move the direction of the force in a more buccal direction, another sectional wire (0.017 × 0.025 TMA) has been directly bonded on the buccal surface of 2.5 and activated, first on buccal direction and then on vertical direction. When using this type of connection, in order to minimize the effect of play between the wire and the slot, it is more biomechanically useful to create a bonded composite connection between the anchorage wire, the adjacent teeth, and the TAD. The procedure is shown of Figure 5. 

#### 2.2.3. Refine Alignment and Occlusion

Torque finishing with aligners is the last stage of the treatment, which requires another refinement stage to obtain proper torque and inclination (Figure 6); in this case, a further stage of 18 aligners was necessary. The final occlusion is shown on Figure 7. The final X-ray analysis is presented on Figure 8.

### 2.3. Case Two

An adult female patient, 43 years old, with Class I occlusion on both sides and with a slight deep bite, light crowding, a persistent deciduous canine no. 63, and a horizontally impacted canine with the crown tip close to the lateral incisors and the tip of the root protruding beyond the buccal bone in the 2.3 area is presented in Figure 9 and Figure 10.

The therapeutic choice was to start with mini-invasive laser gingivectomy (Figure 11) followed immediately by traction by means of TAD direct anchorage. In order to reduce bone resistance, a piezocision cut (Figure 12) along the eruption path of the canine was performed. The biomechanics system for force eruption consisted of a piece of mesh (Leone, Italy), precisely cut to fit the palatal side of the canine, connected to a 150 g coil spring by means of a metallic ligature (Figure 12). The TAD (HDC Spider Pin, 1.3 mm wide and 10 mm long) was inserted on the palatial side mesially to the first molar and then connected to the coil spring. Thus, the extrusion vector was diagonal and horizontal in order to move the crown tip far away from the lateral incisor, and, at the same time, it was vertical in order to produce extrusion (Figure 13).

After the canine eruption, the same TAD was removed from the palatal position and positioned buccally in order to refine the canine position.

The final phase involves using aligners to correct canine torque. It can be achieved with a lingual attachment and then a lingual button and triangular criss/cross elastics and a number of 38 aligners to correct the malocclusion and refine the canine position.

Finalization of wire extrusion (0.016 × 0.022 CuNiTi), combined with aligners not based on TAD anchorage, was planned at the end of the treatment.

Figure 14 presents a set of pictures showing all treatment phases as well as the final teeth position on the dental panoramic radiograph.

## 3. Results

The canine position was fully recovered within approximately two years without any periodontal issues or using conventional brackets in both cases. Final clinical records show good esthetics and functional recovery of upper canines in the arch. The two clinical conditions described in a couple of case reports resulted in two specific protocols, one with two TADs and the other one with a TAD placed first palatally and then buccally.

Of the two presented cases of impacted canine treatment after bite alignment and “canine first approach”, the latter seems to the authors more favourable from the point of view of treatment time and patient’s motivation; tracking the canine before starting treatment with aligners results in shorter treatment time and a lower number of aligners used, but of course it can be applied only when the space for the impacted canine can be fully available.

## 4. Discussion

The traction of palatally impacted canines is one of the greatest challenges of orthodontic treatment, and success depends on many factors: incisor overlap, vertical height, angulation, and apex position. The conventional method of orthodontic-surgical treatment of impacted canines consists in surgical tooth exposing, using the “open exposure” or “closed exposure” method, bonding an orthodontic button with the connected chain intraoperatively, and connecting it to the arch of the fixed appliance.

For many years, the technique of “closed exposure” has been preferred and considered safer for the periodontium. Recent studies show, however, that both methods give similar treatment effects, and the time of tooth insertion into the arch [13] is similar. Moreover, the closed eruption technique induces ankylosis more often (14.5%) than the open eruption technique (3.5%) [14], and all previous studies preferring one of the techniques were based on a small group of patients and low-quality research methods [15,16].

Transferring long-term orthodontic forces directly to the teeth incorporated into the fixed appliance, especially to the adjacent incisors, creates a risk of root resorption [17]. Each year of orthodontic treatment with fixed appliances causes systematic root resorption—0.9 mm each year [18,19]. The exemplary forces bringing the canine into the oral cavity are often more than 20–26 g; they are considered excessive, influencing the ischemia of the periodontium, and if they persist for more than 6-8 weeks, they stimulate the process of pathological root resorption [20]. To avoid this complication, modifications to the anchorage have been introduced. Along with the development of the use of skeletal anchorage in orthodontic treatment, procedures have been introduced to standardize the use of micro-screws as the point of transfer of the pulling force from the retained clause. This method enables faster tooth restoration, precise modification of the direction of bringing it away from the root apexes of the adjacent teeth, as well as avoidance of resorption, which is caused by mechanical damage to the adjacent roots by their contact with the imported tooth and not by applying forces to the adjacent teeth.

Another improvement is an attempt to use aligners for treating impacted teeth in palatal impaction only in conjunction with skeletal anchorage. There are many companies producing aligners on the market today. The first company that developed this treatment method was a medical device company, Align Technology, which produces Invisalign® aligners. The improvements introduced in recent years now allow for the transfer of data and images from CBCT to Clin-Check to visualize the planned course of treatment. Panoramic imaging can help to predict the kind of impaction of maxillary canines, but only CBCT can identify the location of impacted maxillary canines precisely. Precise planning of the transfer of forces to the tooth roots, enabling torque movements, is necessary for the correct completion of bringing the tooth into the arch. The long-term position of the root and periodontal health determine the success of the treatment: the presence or absence of recession, the level of the bone around the root, and the position of the root in the alveolar process. This direction of orthodontics development seems to be obvious, considering the advantages and possibilities it offers. First of all, constant, predictable forces generated by aligners enable the safest periodontal positioning of the teeth, including reduced risk of periodontal ischemia and the associated risk of mechanical damage to the gums during brushing, which is a high risk when using fixed appliances [21]. Other advantages stimulating the increase in popularity of this treatment method among patients and physicians are the patient’s independence from regular and frequent orthodontic visits, reduced pain associated with orthodontic tooth moving, no mechanical damage to the mucosa from the brackets and arches used, improved hygiene possibilities, and, of course, the aesthetics are comparable only to currently less popular lingual braces [22].

The lack of retention elements in aligners makes attaching of the impacted tooth to the elements in oral cavity impossible. Therefore, the use of aligners as a method of choice, in the case of especially palatally-impacted canines, needs a hybrid connection with mini-screws and often also partial arches. Only such a combination makes it possible to take advantage of the benefits of aligners, described above. According to the study by Magliorati et al. comparing the speed of canine traction by using mini-screws and the palatal arch, no significant difference in time was found [23], but slight differences were found between the studied groups with respect to apex displacement and tip displacement.

Using the micro-screws overcomes the limitations of aligners. The precise planning of the TAD insertion enables the application of force exactly in the place needed for the correct tooth retraction, which often requires derotation, correct root torque, and crown tipping. When inserting the micro-screw, the anatomy of the maxilla and mandible should be taken into account; their vascularity and innervation, as well as the bone level to which the length of the microimplant is inserted into contact with the periosteum, should be adjusted. For this reason, the success of treatment is primarily determined by the doctor’s knowledge and skills, but also the method gives the possibility of an individual and personalized look at a specific problem. It can be said that the TADs and the adhesive method, so developed in temporary dentistry, enabled efficient treatment of even the most difficult cases. The only limitation of the described method, apart from possible medical contraindications in a particular patient, is the need to obtain the ideal intraoperative dryness necessary for the effective use of adhesive methods connecting the tooth/teeth with a mini-implant. It also depends on the efficient operation of the operator and assistant, as well as on the developing field of materials science and the selection of the optimal bonding system.

So far, the appliance of choice during the traction procedure of an impacted tooth was a fixed appliance, and previously, additionally, a transpalatal arch (TPA). The advantage of the fixed appliance is the possibility of attaching the pulling element directly from the impacted canine to the arch of the fixed appliance, However, surgical exposure of the canine is still necessary and, in most cases, both operations can be performed in one procedure. Taking into account the advantages of aligners, especially in comparison with fixed braces, in this case primarily facilitating hygiene during the several months and sometimes even many years of treatment and thus avoiding complications in the form of decalcification, caries, inflammatory periodontal diseases, this treatment protocol seems to be method of choice. Additionally, the economic-financial aspect, which seemingly seems to differentiate these two methods, after a deeper analysis, turns out to be not so significant; the use of micro-screws accelerates tracking the impacted tooth to the arch, so the cost of the TAD is comparable to or lower than the sum of the costs of monthly control visits when activating fixed appliances. Similarly, the cost of treatment with aligners is now very much equal to the cost of treatment with fixed braces.

## 5. Conclusions

The use of aligners and TADs seems to be an optimal combination considering the current state of knowledge.The use of aligners in the treatment of impacted canines in combination with TADs and sectional wires can represent a viable alternative option to the conventional systems for canine disinclusion.When the whole treatment is managed with the presented approach following the two different protocols described, no further cooperation with the patient is required in order to support the forced eruption, and an ideal biomechanical approach can be easily applied with one or two mini-screws.

## Author Contributions

The impact of the authors into the paper: M.G.—preparing the figures of the cases, writing the paper, translating; M.M.—writing the paper, translating, supervising. All authors have read and agreed to the published version of the manuscript.

## Figures and Tables

**Figure 1 ijerph-20-00131-f001:**
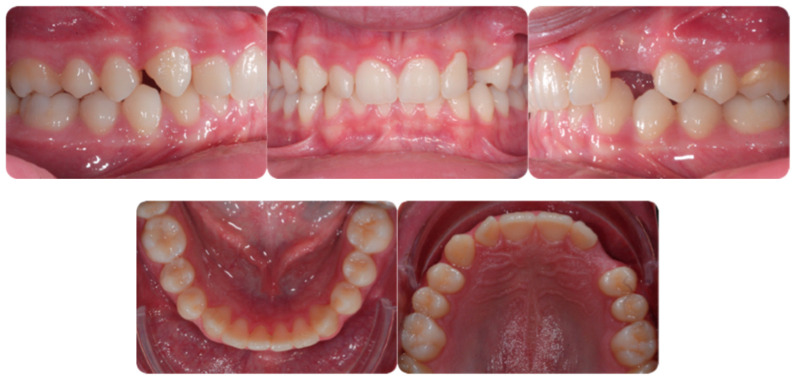
Pre-treatment records of the first patient.

**Figure 2 ijerph-20-00131-f002:**
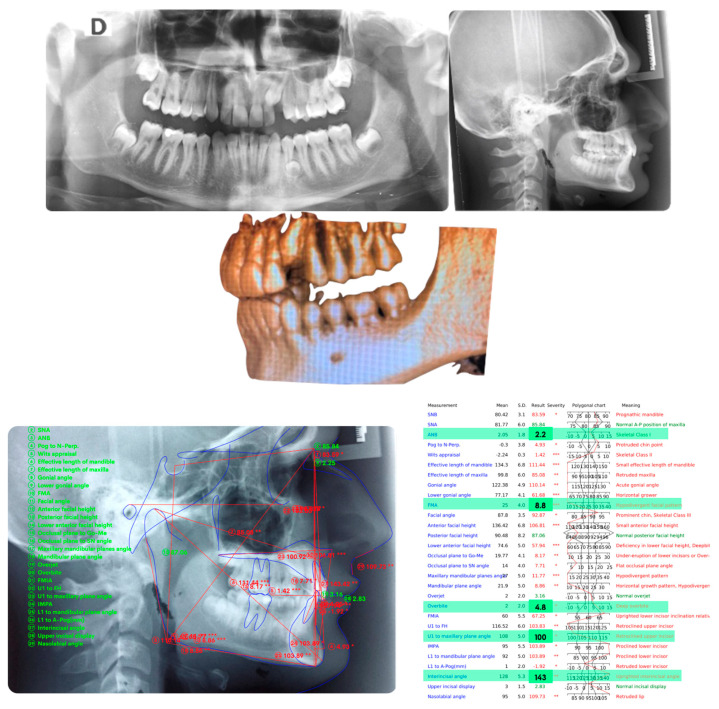
Cephalometric analysis, panoramic and CBCT of the first patient. Lateral X-rays confirming a Class I skeletal and dental malocclusion with deep bite.

**Figure 3 ijerph-20-00131-f003:**
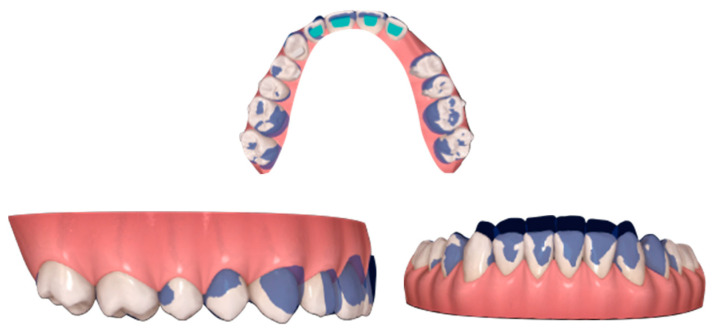
Digital setup planned with expansion and proclination.

**Figure 4 ijerph-20-00131-f004:**
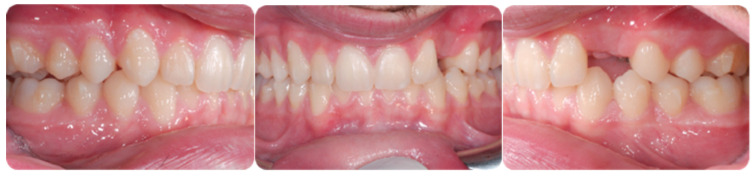
Initial orthodontic vertical traction on a 0-18 SS wire.

**Figure 5 ijerph-20-00131-f005:**
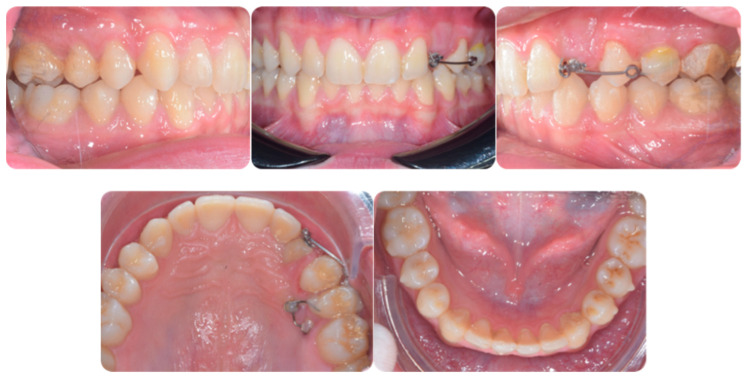
A bonded composite connection between the impacted canine (premolars and premolar) and the mini-screw.

**Figure 6 ijerph-20-00131-f006:**
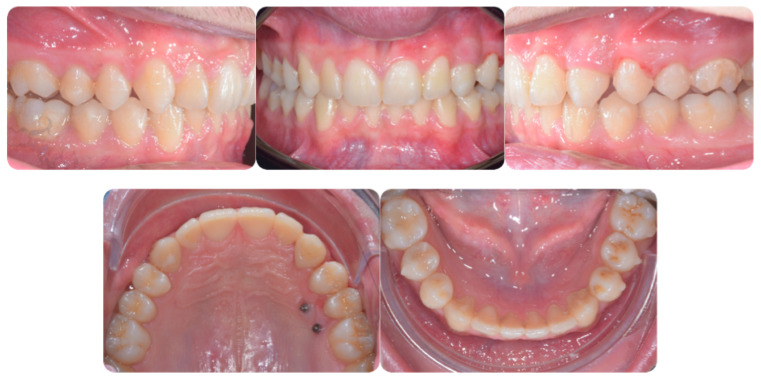
Progress of treatment after canine recovery and before aligner second phase.

**Figure 7 ijerph-20-00131-f007:**
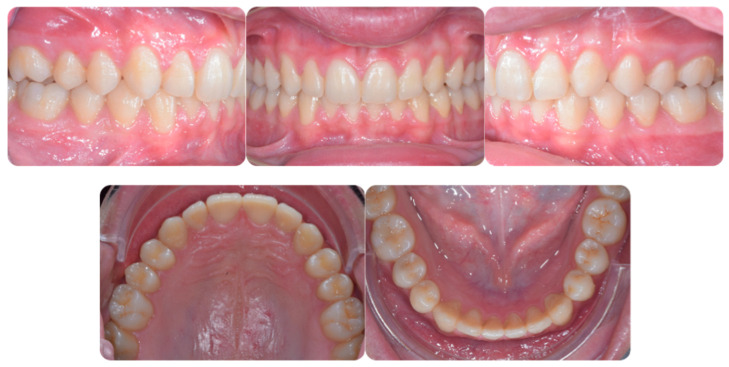
Post-treatment intraoral pictures after 18 months of treatment.

**Figure 8 ijerph-20-00131-f008:**
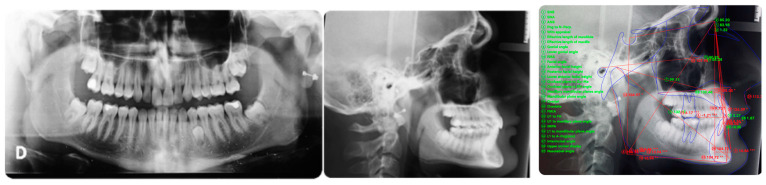
Post-treatment panoramic and lateral X-rays. The supernumerary lower tooth will be extracted following patient need in a second time. Cephalometric superimposition shows the torque correction and deep bite resolution.

**Figure 9 ijerph-20-00131-f009:**
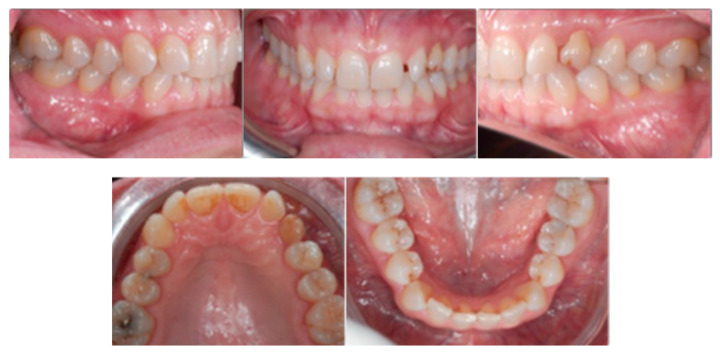
Class I occlusion on both sides with a slight deep bite, light crowding, a persistent deciduous canine no. 63, and a horizontally impacted canine with the tip of the crown close to the lateral incisors and the tip of the root protruding beyond the buccal bone in the 2.3 area.

**Figure 10 ijerph-20-00131-f010:**
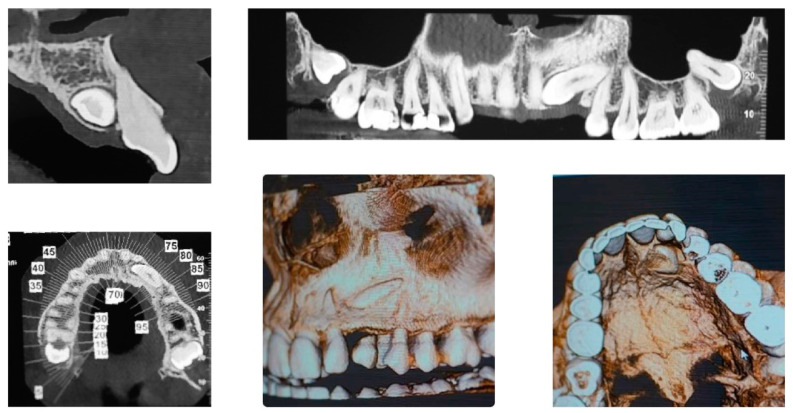
CBCT pictures of the patient. Since the patient was subjected to periodic radiographic control as she was a cancer survivor, the decision was to do a CBCT to receive all the informations useful for the deimpaction treatment and avoid conventional orthodontic X-rays like panoramic and lateral X-rays.

**Figure 11 ijerph-20-00131-f011:**
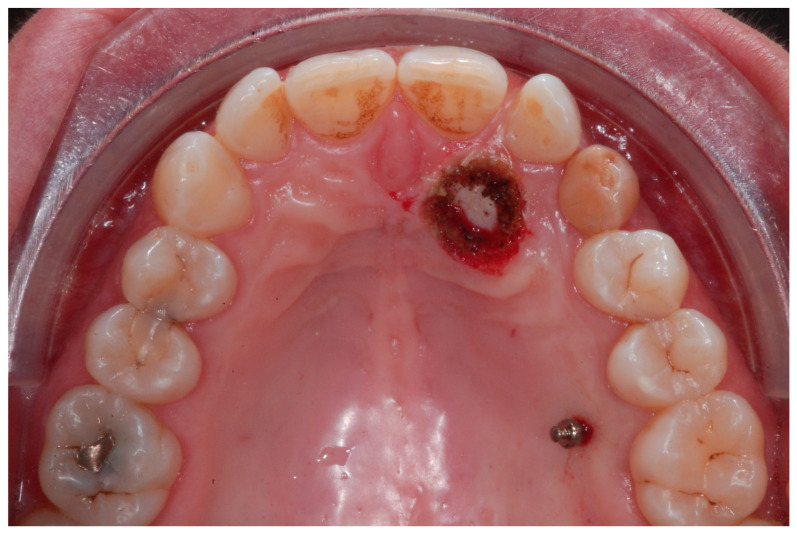
Mini-invasive laser gingivectomy.

**Figure 12 ijerph-20-00131-f012:**
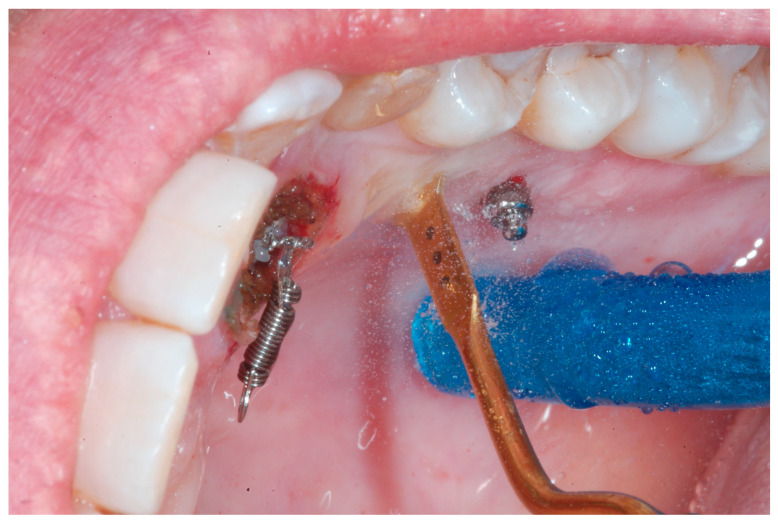
A piezocision cut carried out along the eruption path of the canine in order to reduce bone resistance.

**Figure 13 ijerph-20-00131-f013:**
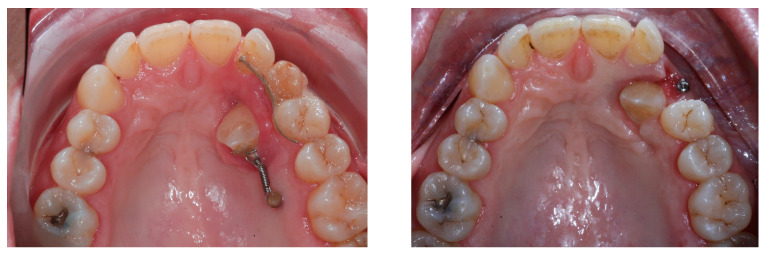
The TAD inserted on the palatial side mesially to the first molar and connected to the coil spring. away from the lateral incisor, and, at the same time, it was vertical in order to produce extrusion.

**Figure 14 ijerph-20-00131-f014:**
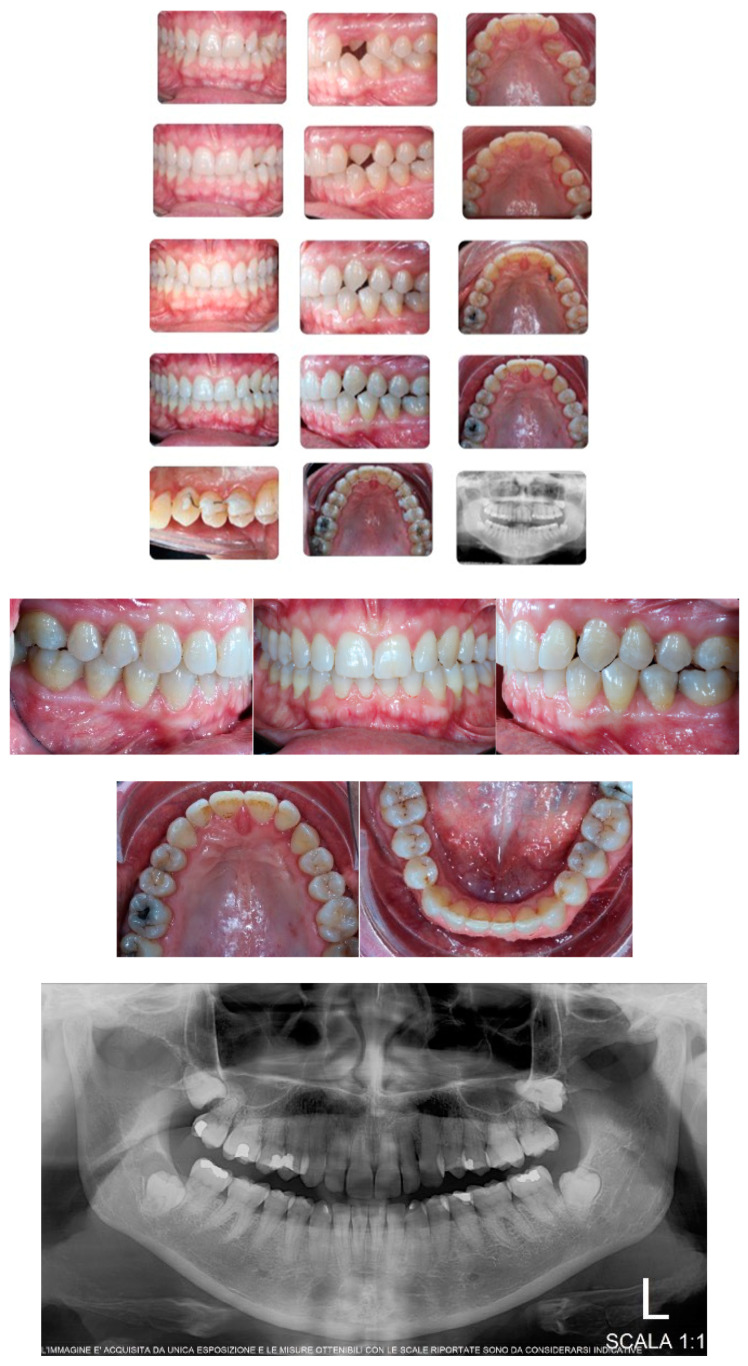
Presentation of all treatment phases as well as the final teeth position on the dental panoramic radiograph. For the same reason described above, only a panoramic X-ray was prescribed to the patient.

## Data Availability

Not applicable.

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
