# Peer review of "Impacted Canine Management Using Aligners Supported by Orthodontic Temporary Anchorage Devices"

_ijerph, 2022, doi:10.3390/ijerph20010131_

Round 1
Reviewer 1 Report
The manuscript entitled “Biomechanical approach of impacted canine management using alingers supported by orthodontic miniscrews – case report” assesses the case presentation about the two ways of treatment with impacted canines. The work is valuable due to the lack of publications on similar topics. Each case report includes a full protocol of the procedure with clarification of purposefulness. The method chosen by the authors is hybrid and use a temporary anchorage device (TAD) and alingers. It is an alternative to classic fixed orthodontic appliances. Despite numerous intraoral and diagnostic photos showing the stages of treatment, I would suggest the correction of descriptions, especially CBCT. In the discussion, I would suggest a summary in the form of a table that would compare the advantages and disadvantages of the hybrid method described by the authors and the method using classic fixed orthodontic appliances.
Author Response
Dear Reviewer,
Thank you very much for the positive and substantive review. We have expanded the cases descriptions and presented in the discussion the advantages of the hybrid method of tracking the impacted teeth. Due to the small number of advantages of fixed appliances, it was difficult for us to present it in the form of a table, but a whole paragraph of discussion was devoted to this problem. We hope that this will be acceptable to the esteemed reviewer.
Yours faithfully,
Authors
Reviewer 2 Report
In this manuscript, you report two clinical cases. However, the text, beginning with your aim to "establish an ideal sequence of orthodontic treatment of impacted canines with aligners, supported by the use of orthodontic miniscrews" reads like a position paper or a consensus report.
A case report needs to be reported like a case report. Thus, the manuscript, especially the style of reporting needs to be throughly revised to reflect the nature of the content. Report both cases in sufficient detail, but refrain from concluding any ideal treatments from just two cases.
Author Response
Dear Reviewer,
According to your points we changed the whole structure of the article starting from the title, we added better documentation and description of the cases.
Thank you for your effort of reviewing the article and giving us very substantive remarks.
Yours faithfully,
Authors
Reviewer 3 Report
This case series does not meet the CARE guidelines.
The figures are unclear.
English is very bad. Even some orthodontic terms have not been translated correctly.
The treatment phase with aligners is only mentioned, but not adequately described.
I would underline that in case 2 the final position of 2.3 seems to be not good since the intercuspation is not perfect and the gingival parable is higher than central incisors and contralateral canine.
The results, discussion and conclusion sections are inappropriate and should be rewritten.
Author Response
Dear Reviewer,
According to your points we changed the whole structure of the article starting from the title, we added better documentation and description of the cases, we improved the form and language, changed the form of conclusion, results, we extended the discussion.
The figures are added and better described, the figures itself- in better quality- will be send to the journal in a separate file.
Thank you for your effort of reviewing the article and giving us very substantive remarks. We hope that after this major changes that we’ve provided it will be more acceptable for you.
Yours faithfully,
Authors